# Diversified Semantic Distribution Matching
# for Dataset Distillation

## ABSTRACT

Dataset distillation, also known as dataset condensation, offers a possibility for compressing a large-scale dataset into a small-scale one (*i.e.*, distilled dataset) while achieving similar performance during model training. This method effectively tackles the challenges of training efficiency and storage cost posed by the large-scale dataset. Existing dataset distillation methods can be categorized into Optimization-Oriented (OO)-based and Distribution-Matching (DM)-based methods. Since OO-based methods require bi-level optimization to alternately optimize the model and the distilled data, they face challenges due to high computational overhead in practical applications. Thus, DM-based methods have emerged as an alternative by aligning the prototypes of the distilled data to those of the original data. Although efficient, these methods overlook the diversity of the distilled data, which will limit the performance of evaluation tasks. In this paper, we propose a novel Diversified Semantic Distribution Matching (DSDM) approach for dataset distillation. To accurately capture semantic features, we first pre-train models for dataset distillation. Subsequently, we estimate the distribution of each category by calculating its prototype and covariance matrix, where the covariance matrix indicates the direction of semantic feature transformations for each category. Then, in addition to the prototypes, the covariance matrices are also matched to obtain more diversity for the distilled data. However, since the distilled data are optimized by multiple pre-trained models, the training process will fluctuate severely. Therefore, we match the distilled data of the current pre-trained model with the historical integrated prototypes. Experimental results demonstrate that our DSDM achieves state-of-the-art results on both image and speech datasets. Codes will be released soon.

## CCS CONCEPTS

• **Computing methodologies → Artificial intelligence**.

## KEYWORDS

Dataset Distillation, Prototype Learning, Semantic Feature

## 1 INTRODUCTION

In the era of big data, the volume of multimedia data grows exponentially, which brings great challenges to storage and training efficiency [12, 33]. For instance, the multi-modal CLIP was trained

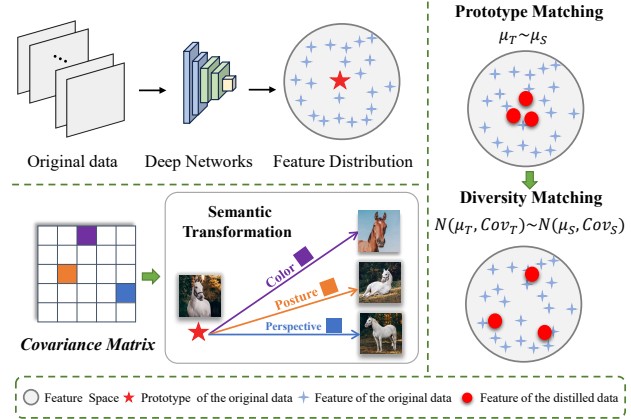

**Figure 1: An illustration of diversified semantic dataset distillation. Unlike previous methods match the prototypes of the original and distilled dataset ($\mu_{\mathcal{T}} \sim \mu_{\mathcal{S}}$), our method matches Gaussian distributions ($\mathcal{N}(\mu_{\mathcal{T}}, Cov_{\mathcal{T}}) \sim \mathcal{N}(\mu_{\mathcal{S}}, Cov_{\mathcal{S}})$).**

on an impressive 400 million data, consuming thousands of GPU days [17, 22, 31]. This poses an interesting issue: how to summarize the large-scale multimedia data into a compact but insightful subset to facilitate the training procedure while maintaining similar performance. To achieve this purpose, an intuitive approach is to select representative instances from the original dataset, known as coreset selection [4, 9, 23, 25, 32]. However, sample selection involves directly discarding a significant portion of instances from the original dataset, leading to the loss of crucial underlying information essential for model training. Fortunately, Dataset Distillation (DD) has been proposed as a solution to condense the original large-scale dataset into smaller ones (*i.e.*, distilled dataset) while ensuring that models trained on the distilled dataset exhibit performance comparable to those trained on the original dataset [6, 16, 30].

Existing methods for dataset distillation can generally be categorized into two main groups: Optimization-Oriented (OO)-based methods and Distribution-Matching (DM)-based methods [36, 42]. OO-based methods employ a bi-level optimization process [2, 8, 30, 41], where the model is first updated with the original dataset, and then, the distilled dataset is optimized by minimizing the distance between the original and distilled data. Although OO-based methods have demonstrated good performance, they are often criticized for their significant time consumption during training. In contrast, DM-based methods are proposed to improve the training efficiency by eliminating model updates [28, 38]. Specifically, DM [40] optimizes the the distilled data by matching corresponding prototypes calculated with a random initialized model on the distilled and original datasets. However, the embeddings generated by a randomly initialized model tend to be inaccurate, which can impact the performance of the distilled dataset. To tackle this issue,

IDM [42] utilizes a sequence of pre-trained models to obtain accurate prototypes. Although these prototypes are more accurate, they ignore the diversity of the distilled dataset, which will reduce the generalization of the models trained on it.

As illustrated in Figure 1, when only prototypes of the distilled data and the original data are aligned to optimize the distilled data, the algorithm may lazily minimize the alignment loss by distilling instances near the center of each class into the distilled data. As a result, all the distilled instances contain similar information, which overlooks the diverse and meaningful semantic information in the original data. Besides the prototype, the covariance matrix is another important statistic to depict the variations of a category, where each item in the covariance matrix reflects the transformation from one-dimensional semantic feature to another. As shown in Figure 1, suppose the "purple, orange, blue" in the covariance matrix represent the transformation from standing to "color transformation, posture transformation, and perspective transformation", applying these transformations to the a white, frontal and standing horse can obtain horses of "brown, lying, and running". Therefore, the covariance matrix carries rich semantic diversity information. Inspired by this, we can improve the diversity of the distilled data by aligning their covariance matrix to that of the original data to make the distilled and original data distribute similarly.

In this paper, we propose a simple yet efficient Diversified Semantic Distribution Matching (DSDM) method for dataset distillation, aiming to obtain semantic diversified distilled data. To acquire accurate semantic information from the original data, we first pre-train a set of models on the original dataset. Then, each category in the original dataset is modeled into a Gaussian distribution with the semantic embedding extracted by a pre-trained model, where the prototype represents the intrinsic feature of a category, and the covariance matrix represents the semantic variations of instances in this category. Subsequently, we align the prototype and covariance matrix of the corresponding distilled data to make the distribution of the distilled data similar to that of the original data. With the additional alignment of the covariance matrix, our distilled data will be distributed dispersedly in the space of the original data rather than gathered around the class centers, ensuring the performance of the distilled data in the evaluation tasks. In addition, since the distilled data are optimized by multiple pre-trained models to make them generalize well to different models, the training process will fluctuate severely. Therefore, we introduce a memory bank to store the integrated prototypes of distilled data optimized by previous pre-trained models. Finally, the distilled data of the current pre-trained model are aligned with the historical integrated prototypes to ensure that the distilled data generalize well across various models. Empirical evaluations conducted on five image datasets and a speech dataset demonstrate that our method not only preserves the efficiency of DM-based approaches but also exhibits significant performance improvements.

The contributions of this paper are summarized as follows:

- We introduce a novel approach called Diversified Semantic Distribution Matching (DSDM) for dataset distillation. To the best of our knowledge, this is the first work to align features from a semantic distribution perspective, thereby enhancing the semantic diversity of distilled dataset.

- We analyze the disparities in feature space distributions among different models. To ensure stable optimization of distillation, we align the prototype of the current distilled data with the historically optimized distilled data.

- We conduct a series of experiments to thoroughly validate the effectiveness of DSDM on both image and speech datasets. Through rigorous experimentation, our DSDM has demonstrated state-of-the-art performance with robust generalization capabilities across various scenarios.

## 2 RELATED WORK

### 2.1 Coreset Selection

The classical approach to compressing the original dataset is coreset selection or instance selection. In addition to random sampling from the original dataset, most existing strategies progressively select important data points based on some heuristic selection criteria. For example, Herding [5, 32] minimizes the distance between the feature space's center of the chosen subset and that of the original dataset by greedily incorporating one instance at a time during each iteration. K-Center [9, 25] selects data points that minimize the maximum distance between a data point and its nearest center. Gradient-Greedy [1] employs gradients as features to maximize the diversity of instances in the replay. However, these heuristic selection criteria do not guarantee that the selected subset is optimal for training the model, particularly in the deep neural networks.

### 2.2 Dataset Distillation

Existing Dataset Distillation methods can be classified into two types: Optimization-Oriented (OO)-based and Distribution Matching (DM)-based methods [36, 42].

**Optimization-Oriented (OO)-based methods** integrate information learned from the dataset into the synthesized dataset during model training, necessitating iterative bi-level optimization of both the model and distilled dataset. Wang et al. are the first to introduce the concept of dataset distillation from an optimization perspective, updating synthesized instances through meta-learning [30]. DC utilizes this bi-level meta-learning optimization approach to match gradients of the distilled dataset and the original dataset to simulate the training process [41]. DSA further improves data efficiency by introducing differentiable Siamese augmentation, enabling effective training of neural networks with augmented data [39]. IDC injects a differentiable multi-formation function into synthesized instances within the constraint of fixed distilled dataset storage [13]. Along these lines, various approaches have been proposed to generate distilled instances, including matching the training trajectories [2], using data hallucination networks to construct sample features [26], and selecting original sample features via clustering [18]. Although these methods have improved performance of the distilled data, they still follow the bi-level optimization paradigm to update the model and the distilled data alternately, which is difficult to be optimized and time-consuming.

**Distribution-Matching (DM)-based methods** directly align the distribution of distilled dataset with that of the real dataset in the feature space [28, 40]. They avoid the bi-level optimization by leaving out the updating of the model. DM [40] is proposed to match the average feature representations of the original dataset

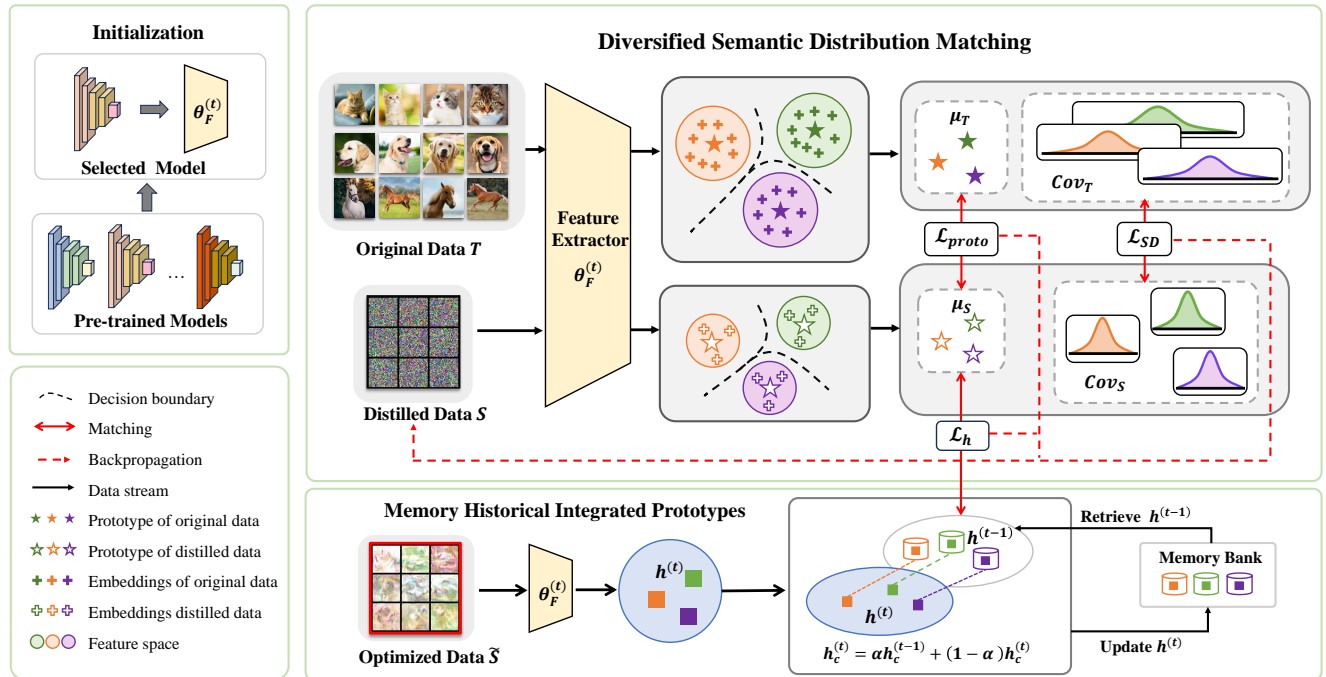

**Figure 2: The illustration of our proposed DSDM. First, DSDM instantiates a feature extractor $\theta_F^{(t)}$ by randomly selecting a model from the pre-trained models. Then, DSDM utilizes three key components to optimize the distilled data: prototype alignment ($\mathcal{L}_{proto}$), semantic diversity alignment ($\mathcal{L}_{SD}$), and historical model alignment ($\mathcal{L}_h$). After the $t$-th iteration of optimization, DSDM updates and stores the historically optimized prototypes $h^{(t)}$ in the memory bank.**

with those of the distilled dataset. However, this approach ignores the discriminative nature of the classes. To address this limitation, CAFE [28] employs a discriminative loss that utilizes the prototypes of the distilled instances as classifiers for the real instances. However, the embeddings to calculate the prototypes are extracted with a randomly initialized model, which are usually inaccurate and will affect the performance of distillation. To address this issue, IDM [42] introduces a queue of pre-trained models to find more accurate class centers. In addition, M3D [36] embeds the class centers into the Reproducing Kernel Hilbert space, which aligns the distributions more efficiently. We recognize that only aligning class centers while overlooking the semantic diversity in the original dataset will produce oversimplified distilled data, which are similar to the class centers. Therefore, our method considers semantic information to enhance the diversity of distilled data.

## 3 PRELIMINARY

Dataset distillation is to condense a large-scale training dataset $\mathcal{T} = \{(x_i, y_i)\}_{i=1}^{|\mathcal{T}|}$ into a small-scale distilled dataset $\mathcal{S} = \{(s_j, y_j)\}_{j=1}^{|\mathcal{S}|}$, where models trained on $\mathcal{S}$ resemble those trained on $\mathcal{T}$. Typically, the distilled dataset $\mathcal{S}$ is initialized with randomly selected original instances from $\mathcal{T}$ or random noise. Then, $\mathcal{S}$ is optimized by minimizing the information loss between the distilled and the original examples, which can be formulated as:

$$\mathcal{S} = \arg\min_{\mathcal{S}} D(\phi(\mathcal{T}), \phi(\mathcal{S})), \tag{1}$$

where $D$ represents a distance metric such as Mean Square Error (MSE), and $\phi$ denotes the matching objective. As mentioned above, based on whether to perform a costly bi-level optimization, existing methods can be mainly divided into Optimization-Oriented (OO)-based and Distribution-Matching (DM)-based methods [35, 36].

In OO-based methods, $\phi$ typically represents the gradients. In each iteration, to simulate different backbones, the parameters of a model are initialized to $\theta^{(0)}$. Then, in the outer loop, $\theta^{(0)}$ is updated for $\hat{N}$ steps using $\mathcal{T}$ to generate gradients through a training procedure. Next, in the inner loop, $\mathcal{S}$ is optimized by minimizing the gradient matching losses between the original and distilled data. The loss function can be defined as follows:

$$\mathcal{S} = \arg\min_{\mathcal{S}} \mathbb{E}_{\theta^{(0)} \sim P_\theta} \left[ \sum_{i=0}^{\hat{N}} D\left( \nabla l\left( \mathcal{S}; \theta^{(i)} \right), \nabla l\left( \mathcal{T}; \theta^{(i)} \right) \right) \right],$$
$$\text{with} \quad \theta^{(i)} = \theta^{(i-1)} - \eta \nabla l\left( \mathcal{T}; \theta^{(i-1)} \right), \quad i \geq 1, \tag{2}$$

where $P_\theta$ is the distribution for the initialization of network parameters, $l$ is the empirical cross-entropy loss and $\eta$ is the learning rate for updating the network.

Although OO-based methods achieve superior performance, these methods incur significant time overhead due to their bi-level loops. To mitigate this challenge, DM-based methods aim to generate distilled data to approximate the original data by directly matching the class centers of the distilled dataset with those of the

original dataset [40]. This objective can be formulated as:

$$\mathcal{S} = \arg\min_{\mathcal{S}} E_{\boldsymbol{\theta} \sim P_{\boldsymbol{\theta}}} \left[ D \left( g(\boldsymbol{\theta}_F; \mathcal{S}), g(\boldsymbol{\theta}_F; \mathcal{T}) \right) \right], \quad (3)$$

where $\boldsymbol{\theta}_F$ denotes the feature sub-network parameterized by $\boldsymbol{\theta}$, instantiated by the model without the output layer and the function $g$ represents the embedding function parameterized with $\boldsymbol{\theta}_F$.

# 4 METHOD

## 4.1 Overview

We illustrate the framework of our proposed Diversified Semantic Distribution Matching (DSDM) in Figure 2. To obtain accurate instance embeddings, we train multiple models using the original dataset. For the $t$-th iteration, we initialize a feature sub-network $\boldsymbol{\theta}_F^{(t)}$ instantiated by randomly selecting one of these pre-trained models. Then, we derive instance embeddings from both $\mathcal{T}$ and $\mathcal{S}$. The extracted features are then used to compute prototypes $\boldsymbol{\mu}_{\mathcal{S}}$ and $\boldsymbol{\mu}_{\mathcal{T}}$, as well as covariance matries $Cov_{\mathcal{S}}$ and $Cov_{\mathcal{T}}$ for each category in both $\mathcal{T}$ and $\mathcal{S}$. The prototypes encapsulate inherent features, while the covariance matrices capture the semantic diversity of all features within each category. We then match the prototypes and covariance matries for each category. However, the feature distributions of $\mathcal{S}$ from different pre-trained models can exhibit significant discrepancies during the iteration process, as elaborated in Section 4.3. To mitigate the instability stemming from these differences, we align the historical prototypes of distilled data $\boldsymbol{h}^{(t-1)}$ with the current distilled data $\boldsymbol{\mu}_{\mathcal{S}}$. To preserve the historical prototypes for each pre-trained model, we establish a memory bank to memory historical integrated prototypes. Finally, we summarize the total loss to optimize distilled instances.

## 4.2 Diversified Semantic Distribution Matching

In the deep neural networks, disparate instances belonging to the same category often undergo nuanced semantic transformations within their deeper feature representations [3, 27, 37]. Given that covariance matrices encapsulate both the interrelationships and variances among features, they provide a rich understanding of semantic characteristics. We argue that leveraging category-specific covariance matrices enables the representation of features with distinctive semantic transformations. Thus, we derive semantic directions from estimated covariance matrices, assuming a class-centered normal distribution. Our objective simplifies to matching covariance matrices of features generated for each category $\mathcal{S}$ to closely approximate those of $\mathcal{T}$ (i.e., $\mathcal{N}(\boldsymbol{\mu}_{\mathcal{S}}, Cov_{\mathcal{S}}) \sim \mathcal{N}(\boldsymbol{\mu}_{\mathcal{T}}, Cov_{\mathcal{T}})$).

Previous DM [40] utilizes initialized neural networks to measure the Maximum Mean Discrepancy (MMD) between the original dataset $\mathcal{T}$ and the distilled dataset $\mathcal{S}$. However, aligning the embeddings with randomly initialized neural networks will lead to inaccurate embeddings, as neural networks are ignorant to the current task. Only accurate embeddings can capture effective semantic features. To mitigate this issue, we adopt a strategy of pre-training several models using $\mathcal{T}$. At each iteration $t$, we randomly select one of these pre-trained models and utilize its last layer (excluding the output layer) as the feature extractor $\boldsymbol{\theta}_F^{(t)}$.

For a large-scale original dataset $\mathcal{T} = \{x_i, y_i\}_{i=1}^{|\mathcal{T}|}$, where $y_i \in L = \{l_1, l_2, ..., l_C\}$ and $C$ represents the number of categories, we compress IPC (Instances Per Class) instances for each category $l_c$ (i.e., for the distilled synthetic dataset $\mathcal{S} = \{(s_j, y_j)\}_{j=1}^{|\mathcal{S}|}$, $|\mathcal{S}| = IPC \times C$). $\mathcal{S}$ is usually initialized with instances randomly selected instances from the original dataset. To effectively capture the semantic features of the original dataset $\mathcal{T}$, we use $\boldsymbol{\theta}_F^{(t)}$ to get instance embeddings. We employ the embedding function $g$ to generate embeddings for each instance in both the distilled and original datasets, denoted by $f_i = g(\boldsymbol{\theta}_F^{(t)}; x_i)$ and $\hat{f}_j = g(\boldsymbol{\theta}_F^{(t)}; s_j)$, where $f_i, \hat{f}_j \in R^d$.

Given the inherent differences in the direction of semantic features across classes, we should first align the inherent semantic features of each class between $\mathcal{T}$ and $\mathcal{S}$. Thus, we utilize the euclidean distance to align the prototype of $\mathcal{T}$, denoted as $\boldsymbol{\mu}_{\mathcal{T},c}$, with the prototype of $\mathcal{S}$, denoted as $\boldsymbol{\mu}_{\mathcal{S},c}$. The loss function for this prototype alignment is defined as follows:

$$\mathcal{L}_{proto} = \sum_{c=1}^{C} \|\boldsymbol{\mu}_{\mathcal{T},c} - \boldsymbol{\mu}_{\mathcal{S},c}\|^2$$

$$s.t. \quad \boldsymbol{\mu}_{\mathcal{T},c} = \frac{1}{|B_c^{\mathcal{T}}|} \sum_{i=1}^{|B_c^{\mathcal{T}}|} f_i, \quad \boldsymbol{\mu}_{\mathcal{S},c} = \frac{1}{|B_c^{\mathcal{S}}|} \sum_{j=1}^{|B_c^{\mathcal{S}}|} \hat{f}_j, \quad (4)$$

where $B_c^{\mathcal{T}}$, $B_c^{\mathcal{S}}$ represent a batch of data sampled from the same category $l_c$ in $\mathcal{T}$ and $\mathcal{S}$.

When aligning the prototypes, we let the covariance matrices align to capture semantic diversity. The objective of semantic diversity alignment can be expressed as follows:

$$\mathcal{L}_{SD} = \sum_{c=1}^{C} \frac{1}{d} \left\| Cov_{\mathcal{T},c} - Cov_{\mathcal{S},c} \right\|^2$$

$$s.t. \quad Cov_{\mathcal{T},c}(m,n) = \frac{\sum_{i=1}^{|B_c^{\mathcal{T}}|} \left( f_i^m - \mu_{\mathcal{T},c}^m \right) \left( f_i^n - \mu_{\mathcal{T},c}^n \right)}{|B_c^{\mathcal{T}}|},$$

$$Cov_{\mathcal{S},c}(m,n) = \frac{\sum_{j=1}^{|B_c^{\mathcal{S}}|} \left( \hat{f}_j^m - \mu_{\mathcal{S},c}^m \right) \left( \hat{f}_j^n - \mu_{\mathcal{S},c}^n \right)}{|B_c^{\mathcal{S}}|}, \quad (5)$$

where $(m, n)$ denotes the position of the covariance matrix.

## 4.3 Align Historical Prototypes

To enhance model generalization of distilled data across diverse evaluation tasks, we adopt multiple pre-trained models to capture various semantic features. However, notable differences emerge in the feature distributions of distilled data for the same class across different pre-trained models. In Figure 3 (a), we visualize the feature distributions of the distilled data from a category across six pre-training models. It is obvious that data distilled by different pre-trained models are distributed differently. Figure 3 (b) demonstrates the matching loss between the distilled data and the original data under these six pre-trained models. We can see that the distance between the same distilled data and the same original data varies greatly among different pre-trained models, resulting that training loss for data distillation fluctuations severely when the model for feature extraction changes.

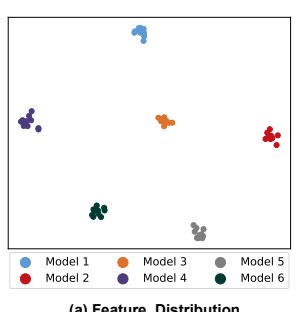
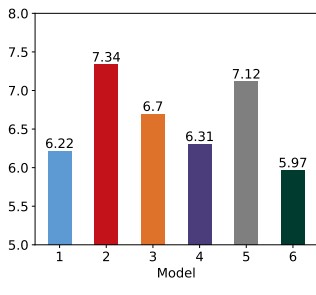

(a) Feature Distribution

(b) Feature Matching Loss

**Figure 3: Illustration of feature distributions and matching loss on the same distilled data under six pre-trained models.**

To enhance the stability of distillation performance over multiple pre-trained models, we introduce a memory bank to store the integrated prototypes of distilled data optimized by previous pre-trained models. Then, we align the distilled data of the current model with the corresponding integrated prototype $h_c^{(t-1)}$. Thus, we compute the historical prototype alignment loss as:

$$\mathcal{L}_h = \sum_{c=1}^{C} \|\boldsymbol{\mu}_{\mathcal{S},c} - \boldsymbol{h}_c^{(t-1)}\|^2. \qquad (6)$$

To store and update the prototypes used for historical prototype alignment, we first compute prototypes $\boldsymbol{h}_c^{(t)}$ for each class after the $t$-th optimization with the corresponding feature extractor $\boldsymbol{\theta}_F^{(t)}$ on the optimized distilled data $\tilde{\mathcal{S}}$:

$$\boldsymbol{h}_c^{(t)} = \frac{1}{|\tilde{\mathcal{S}}_c|} \sum_{\tilde{s}_k \in \tilde{\mathcal{S}}_c} g(\boldsymbol{\theta}_F^{(t)}; \tilde{s}_k), \qquad (7)$$

where $\tilde{\mathcal{S}}_c$ represents the distilled data of category $l_c$ after the $t$-th optimization. Next, we update the integrated prototypes in the memory bank:

$$\boldsymbol{h}_c^{(t)} = \begin{cases} \alpha \boldsymbol{h}_c^{(t-1)} + (1-\alpha)\boldsymbol{h}_c^{(t)}, & t > 1 \\ 0, & t = 1, \end{cases} \qquad (8)$$

where $\alpha$ is a smoothing factor, typically set to be 0.99.

### 4.4 Overall Loss and Training Algorithm

In a nutshell, we minimize the sum of the above losses, including the prototype alignment loss $\mathcal{L}_{proto}$, semantic diversity alignment loss $\mathcal{L}_{SD}$, and the historical prototype alignment loss $\mathcal{L}_h$. Combining these loss terms together, our diversified semantic distribution matching loss can be formulated as:

$$\mathcal{L}_{total} = \mathcal{L}_{proto} + \lambda_1 \mathcal{L}_{SD} + \lambda_2 \mathcal{L}_h, \qquad (9)$$

where $\lambda_1$ and $\lambda_2$ are hyperparameters for balancing these terms.

The pseudocode for DSDM is provided in Algorithm 1. In addition, we adopt partitioning and expansion augmentation for the initialization of $\mathcal{S}$, as proposed in IDM [42], aims to increase the number of representations extracted from $\mathcal{S}$ without incurring additional storage costs. Specifically, employing a factor parameter $l$, each distilled instance is factorized into $l \times l$ mini-examples, which are then up-sampled to their original size during training. This

---

**Algorithm 1: DSDM**

**Input:** $\mathcal{T}$: original training dataset; $\mathcal{S}$: distilled synthetic dataset; $T$: number of training iterations; $C$: number of class; $\eta_{DSDM}$: learning rate of optimizing distilled data; $M$: number of pre-trained networks.

1 Randomly train $M$ models $\{\theta_m\}_{m=1}^M$ with $\mathcal{T}$;
2 Initialize distilled dataset $\mathcal{S}$;
3 **for** $t = 1 \leftarrow \boldsymbol{to}\ T$ **do**
4      Randomly load a checkpoint from the pre-trained models to instantiate a feature extractor $\boldsymbol{\theta}_F^{(t)}$.
5      Sample an intra-class mini-batch $B_c^{\mathcal{T}} \sim \mathcal{T}, B_c^{\mathcal{S}} \sim \mathcal{S}$ for each category $c$ in $C$;
6      Compute prototype matching loss $\mathcal{L}_{proto}$ with Eq.(4).
7      Compute semantic diversity matching loss $\mathcal{L}_{SD}$ with Eq.(5).
8      Compute historical prototype alignment loss $\mathcal{L}_h$ with Eq.(6).
9      Compute the total loss $\mathcal{L}_{total}$ with Eq.(9).
10      Update distilled data $\mathcal{S}$: $\mathcal{S} \leftarrow \mathcal{S} - \eta_{DSDM}\nabla_{\mathcal{S}}\mathcal{L}_{total}$
11      Compute and update the historical prototype of the optimized data $\boldsymbol{h}_c^{(t)}$ for each category $c$.
12      Store $\boldsymbol{h}^{(t)} = \{\boldsymbol{h}_c^{(t)}\}_{c=1}^C$ into the memory bank.

**Output:** Distilled synthetic dataset $\mathcal{S}$.

---

approach effectively maximizes the utilization of storage space in $\mathcal{S}$. We integrate the same augmentation technique into our work, thereby further leveraging its advantages in aligning distributions for enhanced diversity.

## 5 EXPERIMENTS

### 5.1 Experimental Setup

**Datasets.** To evaluate the effectiveness of our dataset distillation method, we conduct experiments on five image datasets and one speech dataset. The five image datasets are MNIST [15], Fashion-MNIST (FMNIST) [34], SVHN [19], CIFAR-10 [14], and CIFAR-100 [14]. Specifically, MNIST is a black and white dataset consisting of 60,000 training images and 10,000 test images from 10 different classes. SVHN is a color dataset and consists of 73,257 digits and 26,032 digits for training and testing respectively. CIFAR-10/100 consists of 50,000 training images and 10,000 test images from 10 and 100 different classes respectively. For the speech dataset, we use the Mini-Speech-Commands dataset [7], consisting of 8000 one-second audio clips representing eight command categories.

**Implementation Details.** We follow the configuration of DM [40] to implement our method. Specifically, we use a 3-layer convolutional network (ConvNet-3) and a 4-layer convolutional network (ConvNet-4) [24] for the image and speech datasets, respectively. We initialize $M = 10$ pre-trained models, each trained for $E = 20$ epochs. We execute $T = 10K$ matching optimization iterations. For the weight parameters, we set $\lambda_1 = 50$ and $\lambda_2 = 0.2$ (see section 5.5 for further analysis). Moreover, we set the optimization learning rate for distilled instances $\eta_{DSDM} = 0.1$. Following IDM [42], we fix the factor parameter $l$ to 2 across all datasets. We use Top-1 test

**Table 1: Top-1 accuracy of test models trained on distilled synthetic images on multiple datasets. We evaluate our method on five different datasets with different numbers of synthetic images per class. Ratio(%): the ratio of distilled images to the whole training set. Whole: the accuracy of the model trained on the whole original training set. Note: DD$^{\dagger}$ uses different architectures i.e. LeNet for MNIST and AlexNet for CIFAR10. The other methods use ConvNet-3.**

| Datasets | MNIST | | | FashionMNIST | | | SVHN | | | CIFAR-10 | | | CIFAR-100 | | |
|---|---|---|---|---|---|---|---|---|---|---|---|---|---|---|---|
| IPC | 1 | 10 | 50 | 1 | 10 | 50 | 1 | 10 | 50 | 1 | 10 | 50 | 1 | 10 | 50 |
| Ratio (%) | 0.017 | 0.17 | 0.83 | 0.017 | 0.17 | 0.83 | 0.014 | 0.14 | 0.7 | 0.02 | 0.2 | 1 | 0.2 | 02 | 10 |
| Whole | 99.6$_{\pm0.0}$ | | | 93.5$_{\pm0.1}$ | | | 95.4$_{\pm0.1}$ | | | 84.8$_{\pm0.1}$ | | | 56.2$_{\pm0.3}$ | | |
| Random | 64.9$_{\pm3.5}$ | 95.1$_{\pm0.9}$ | 97.9$_{\pm0.5}$ | 51.4$_{\pm3.8}$ | 73.8$_{\pm0.7}$ | 82.5$_{\pm0.7}$ | 14.6$_{\pm1.6}$ | 35.1$_{\pm4.1}$ | 70.9$_{\pm0.9}$ | 14.4$_{\pm2.0}$ | 26.0$_{\pm1.2}$ | 43.4$_{\pm1.0}$ | 1.4$_{\pm0.1}$ | 5.0$_{\pm0.2}$ | 15.0$_{\pm0.4}$ |
| Herding | 89.2$_{\pm1.6}$ | 93.7$_{\pm0.3}$ | 94.8$_{\pm0.3}$ | 67.0$_{\pm1.9}$ | 71.1$_{\pm0.7}$ | 71.9$_{\pm0.8}$ | 20.9$_{\pm1.3}$ | 50.5$_{\pm3.3}$ | 72.6$_{\pm0.8}$ | 21.5$_{\pm1.2}$ | 31.6$_{\pm0.7}$ | 40.4$_{\pm0.6}$ | 8.4$_{\pm0.3}$ | 17.3$_{\pm0.3}$ | 33.7$_{\pm0.5}$ |
| K-center | 89.3$_{\pm1.5}$ | 84.4$_{\pm1.7}$ | 97.4$_{\pm0.3}$ | 66.9$_{\pm1.8}$ | 54.7$_{\pm1.5}$ | 68.3$_{\pm0.8}$ | 21.0$_{\pm1.5}$ | 14.0$_{\pm1.3}$ | 20.1$_{\pm1.4}$ | 21.5$_{\pm1.3}$ | 14.7$_{\pm0.7}$ | 27.0$_{\pm1.4}$ | 8.3$_{\pm0.3}$ | 7.1$_{\pm0.2}$ | 30.5$_{\pm0.3}$ |
| DD$^{\dagger}$ [30] | - | 79.5$_{\pm8.1}$ | - | - | - | - | - | - | - | - | 36.8$_{\pm1.2}$ | - | - | - | - |
| DC [41] | 91.7$_{\pm05}$ | 97.4$_{\pm0.3}$ | 98.8$_{\pm0.2}$ | 70.5$_{\pm0.6}$ | 82.3$_{\pm0.4}$ | 83.6$_{\pm0.4}$ | 31.2$_{\pm1.4}$ | 76.1$_{\pm0.6}$ | 82.3$_{\pm0.3}$ | 28.3$_{\pm0.5}$ | 44.9$_{\pm0.5}$ | 53.9$_{\pm0.5}$ | 12.8$_{\pm0.3}$ | 25.2$_{\pm0.3}$ | - |
| DSA [39] | 88.7$_{\pm0.6}$ | 97.8$_{\pm0.1}$ | 99.2$_{\pm0.1}$ | 70.6$_{\pm0.6}$ | 84.6$_{\pm0.3}$ | 88.7$_{\pm0.2}$ | 27.5$_{\pm1.4}$ | 79.2$_{\pm0.5}$ | 84.4$_{\pm0.4}$ | 28.8$_{\pm0.7}$ | 52.1$_{\pm0.5}$ | 60.6$_{\pm0.5}$ | 13.9$_{\pm0.3}$ | 32.3$_{\pm0.3}$ | 42.8$_{\pm0.4}$ |
| DM [40] | 89.7$_{\pm0.6}$ | 97.5$_{\pm0.1}$ | 98.6$_{\pm0.1}$ | 70.7$_{\pm0.6}$ | 83.5$_{\pm0.3}$ | 88.1$_{\pm0.6}$ | 30.3$_{\pm0.1}$ | 73.5$_{\pm0.5}$ | 82.0$_{\pm0.8}$ | 26.0$_{\pm0.8}$ | 48.9$_{\pm0.6}$ | 63.0$_{\pm0.4}$ | 11.4$_{\pm0.3}$ | 29.7$_{\pm0.3}$ | 43.6$_{\pm0.4}$ |
| CAFE [29] | 93.1$_{\pm03}$ | 97.2$_{\pm0.2}$ | 98.6$_{\pm0.2}$ | 77.1$_{\pm0.9}$ | 83.0$_{\pm0.4}$ | 84.8$_{\pm0.4}$ | 42.6$_{\pm3.3}$ | 75.9$_{\pm0.6}$ | 81.3$_{\pm0.3}$ | 30.3$_{\pm1.1}$ | 46.3$_{\pm0.6}$ | 55.5$_{\pm0.6}$ | 12.9$_{\pm0.3}$ | 27.8$_{\pm0.3}$ | 37.9$_{\pm0.3}$ |
| CAFE+DSA [29] | 90.8$_{\pm0.5}$ | 97.5$_{\pm0.1}$ | 98.9$_{\pm0.2}$ | 73.7$_{\pm0.7}$ | 83.0$_{\pm0.3}$ | 88.2$_{\pm0.4}$ | 42.9$_{\pm3.0}$ | 77.9$_{\pm0.6}$ | 82.3$_{\pm0.4}$ | 31.6$_{\pm0.8}$ | 50.9$_{\pm0.5}$ | 62.3$_{\pm0.4}$ | 14.0$_{\pm0.3}$ | 31.5$_{\pm0.2}$ | 42.9$_{\pm0.2}$ |
| IDM [42] | - | - | - | - | - | - | - | - | - | 45.6$_{\pm0.7}$ | 58.6$_{\pm0.1}$ | 67.5$_{\pm0.1}$ | 20.1$_{\pm0.3}$ | 45.1$_{\pm0.1}$ | 50.0$_{\pm0.2}$ |
| M3D [36] | 94.4$_{\pm0.2}$ | 97.6$_{\pm0.1}$ | 98.2$_{\pm0.2}$ | 80.7$_{\pm0.3}$ | 85.0$_{\pm0.1}$ | 86.2$_{\pm0.3}$ | 62.8$_{\pm0.5}$ | 83.3$_{\pm0.7}$ | 89.0$_{\pm0.2}$ | 45.3$_{\pm0.3}$ | 63.5$_{\pm0.2}$ | 69.9$_{\pm0.5}$ | 26.2$_{\pm0.3}$ | 42.4$_{\pm0.2}$ | 50.9$_{\pm0.7}$ |
| **DSDM** | **94.8$_{\pm0.2}$** | **98.5$_{\pm0.2}$** | **99.2$_{\pm0.1}$** | **80.8$_{\pm0.4}$** | **87.3$_{\pm0.3}$** | **88.9$_{\pm0.3}$** | 60.2$_{\pm0.2}$ | **85.4$_{\pm0.3}$** | **91.3$_{\pm0.2}$** | **45.0$_{\pm0.4}$** | **66.5$_{\pm0.3}$** | **75.8$_{\pm0.3}$** | 19.5$_{\pm0.2}$ | **46.2$_{\pm0.3}$** | **54.0$_{\pm0.2}$** |

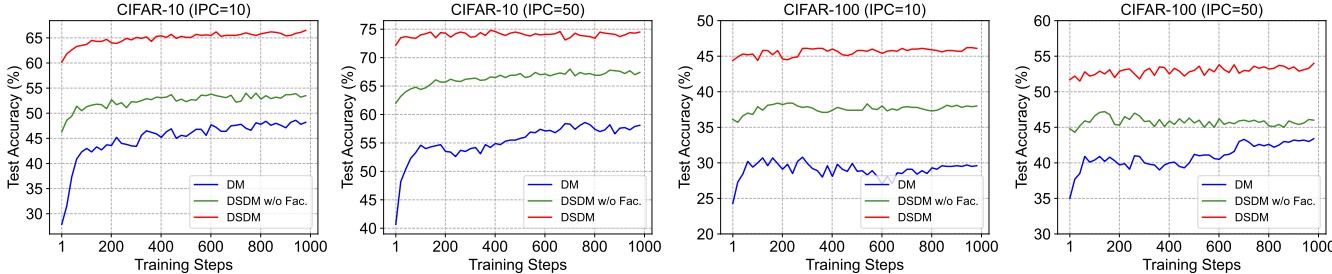

**Figure 4: Performance comparison between DSDM and DM across varying training steps. DSDM w/o Fac. denotes the DSDM without using the factor technique.**

accuracy on the test set of the original dataset as the evaluation metric to assess the performance of a network trained on the distilled dataset. The evaluation task is trained for 1500 epochs. We perform 5 experiments and report the mean and standard deviation of the results.

**Compared Methods.** We compare our DSDM with several state-of-the-art baselines across three categories: Coreset Selection, Optimization-Oriented (OO)-based methods, and Distribution Matching (DM)-based methods. For coreset selection methods, we consider the following: Random [4, 23], Herding [32] and K-Center [9, 25]. For OO-based methods, we include: Dataset Distillation (DD) [30], Dataset Condensation (DC) [41] and DSA (adding differentiable Siamese augmentation for DC) [39]. Although recent OO-based methods [18, 20, 21] achieve better performance, we do not compare them due to the remarkable difference in training scheme and cost. For DM-based methods, we include: CAFE [29], its variant CAFE+DSA [29], DM [40], IDM [42], and M3D [36].

## 5.2 Results on Image Datasets

The performance of our DSDM and that of our competitors on the five image datasets is shown in Table 1. From these results we can draw the following conclusions: Firstly, our DSDM consistently outperforms current state-of-the-art techniques, especially when the IPC in the synthetic dataset is set to 10 and 50. In particular, our method shows significant improvements over the suboptimal method, M3D, by 3.0% and 5.9% on CIFAR-10, respectively. Secondly, using the original dataset (Whole) for training represents the upper bound performance for dataset distillation. Encouragingly, we observe a marginal degradation of only 4.1% at a condensation ratio of 0.7% on SVHN, and a slight decrease of 2.2% at a condensation ratio of 10% on CIFAR-100. These results suggest that our method has the potential to achieve lossless compression by maintaining semantic diversity. Thirdly, the improvements of Our DSDM are limited when IPC=1. This is because the covariance matrix of only one distilled instance cannot be estimated. However, our DSDM

**Table 2: Top-1 accuracy of ConvNet-4 trained on distilled spectrograms.**

| IPC | Method | | | | |
|---|---|---|---|---|---|
| | Random | Herding | DM | DSA | DSDM |
| 10 | 42.6 | 56.2 | 69.1 | 65.0 | 71.1 |
| 20 | 57.0 | 72.9 | 77.0 | 74.0 | 78.9 |

.

**Table 3: Cross-architecture generalization performance (%) on CIFAR-10. The distilled synthetic images is condensed using ConvNet-3 and evaluated using other architectures.**

| IPC | Evalueation | Method | | | |
|---|---|---|---|---|---|
| | | DSA | DM | M3D | Ours |
| 10 | ConvNet-3 | 52.1 | 48.9 | 63.5 | 66.5 |
| | ResNet-10 | 32.9 | 42.3 | 56.7 | 63.7 |
| | DenseNet-121 | 52.1 | 48.9 | 63.5 | 63.8 |
| 50 | ConvNet-3 | 60.6 | 63.0 | 69.9 | 75.8 |
| | ResNet-10 | 49.7 | 58.6 | 66.6 | 70.8 |
| | DenseNet-121 | 49.1 | 57.4 | 66.1 | 68.7 |

still outperforms most of the other methods, because our DSDM aligns features from multiple models to stabilize the training. To further illustrate it, we show the performance of different training stages on CIFAR-10 (IPC=10/50) in Figure 2. It can be observed that our DSDM is more stable and significantly outperforms the compared methods at different training stages.

## 5.3 Results on Speech Dataset

Following the setting in previous work [13], each speech data is pre-processed to be a magnitude spectrogram with the size of 64×64. For the factor parameter $l$, each spectrogram is factorized into 2 mini-examples along the time axis. Table 2 shows the test accuracy for mini-speech commands. It can be seen that our DSDM also greatly outperforms the baseline method, confirming its effectiveness in the speech domain. This means that our approach can be useful in both the image and the speech domain.

## 5.4 Cross-Architecture Evaluation

A more important issue for distilled data is whether they can generalize well to different networks in the evaluation tasks. In Table 3 we show the performance of our distilled dataset (optimized on ConvNet-3) on the evaluation task trained with ConvNet-3, ResNet-10 [10] and DenseNet-121 [11]. Notably, our DSDM not only outperforms other methods on specific distillation model (ConvNet-3) but also shows significant performance gains on other unseen network architectures (ResNet-10 and DenseNet-121). This robust cross-architecture generalization suggests that our distilled dataset with enhanced semantic diversity and integrated information from different pre-models can generalize well to different backbones in the evaluation tasks.

**Table 4: Ablation study of different components in DSDM on CIFAR10 with IPC=10/50.**

| $\mathcal{L}_{proto}$ | PM | $\mathcal{L}_{SD}$ | $\mathcal{L}_h$ | IPC = 10 | IPC = 50 |
|---|---|---|---|---|---|
| ✓ | | | | 60.6 | 68.8 |
| ✓ | ✓ | | | 63.5 | 70.2 |
| ✓ | ✓ | ✓ | | 66.2 | 75.1 |
| ✓ | ✓ | ✓ | ✓ | 66.5 | 75.8 |

## 5.5 Analysis

**Ablation Study.** To evaluate the effectiveness of each component in our DSDM, we conduct ablation experiments on CIFAR-10, including the prototype alignment loss ($\mathcal{L}_{proto}$), pre-trained models (PM), semantic diversity alignment loss ($\mathcal{L}_{SD}$), and historical prototype alignment loss ($\mathcal{L}_h$). The results are presented in Table 4, we can draw the following conclusions: (1) Prototype alignment utilizing pre-trained models exhibits enhancements of 2.9% and 1.4% for IPC=10 and IPC=50, respectively. This suggests that leveraging pre-trained models enables the acquisition of accurate instance embeddings, thereby generating precise intrinsic semantic features for each class. (2) The inclusion of our proposed semantic diversity alignment yields improvements of 2.4% and 4.9% respectively. This underscores the effectiveness of our covariance matrix in capturing semantic diversity. Notably, with the increased IPC (i.e., IPC=50), more diversified distilled instances are obtained, resulting in a greater improvement. (3) Historical prototype alignment emphasizes the capacity to maintain the semantic diversity of feature distributions across various models, it can further boost performance. In summary, the three designed loss functions are complementary and together improve the overall performance.

**Effect of Pre-trained Epochs.** The epochs $E$ for pre-training directly affect the accuracy of instance embeddings (further affect the prototypes and semantic diversity for distillation) extracted by the pre-trained models. To show their effects, we pre-train ConvNets on CIFAR-10 by $E$ in {1, 5, 10, 15, 20, 25, 30} and utilize these pre-trained models to perform our DSDM for IPC=50. As shown in the left of Figure 5, there is a gradual improvement in performance with increasing epochs until reaching optimal performance at $E = 20$. After that, increasing epochs do not significantly affect the performance, indicating that our DSDM only needs a few additional pre-training epochs to acquire better performance.

**Effect of Number of Pre-trained Models.** More pre-trained models mean the distilled data generalize better for the evaluation task. However, more pre-trained models also occupy more training time. Therefore, we perform our DSDM with different numbers of models $M \in \{1, 5, 10, 20\}$ with the pre-training epochs $E = 20$. As illustrated in the right of Figure 5, we show the distillation performance. When M is set to 1 and 5, the performance is slightly worse. This indicates that the number of models has a significant effect on the distillation performance. Despite this, our DSDM achieves optimal performance with only 10 pre-trained models with little storage and computation consumption.

**Effect of $\lambda_1$.** The parameter $\lambda_1$ is a weighting parameter for the degree of improved semantic diversity. Figure 6 illustrates the

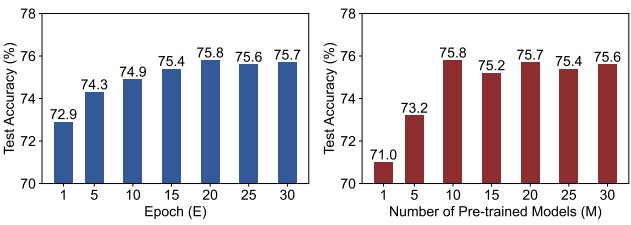

Figure 5: The distillation performance for different pre-train epochs (Left) and different number of pre-trained models (Right). The evaluation is on CIFAR-10 with IPC = 50.

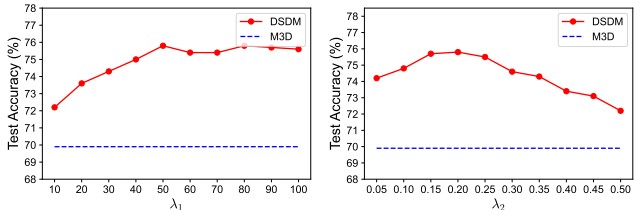

Figure 6: Effect of $\lambda_1$ and $\lambda_2$ on CIFAR-10 with IPC=50.

effect of $\lambda_1$ on the distillation performance for IPC=50 on CIFAR-10. We find that performance improves as the range of $\lambda_1$ goes from 10 to 50, while the effect of increasing the range of $\lambda_1$ from 50 to 100 is almost negligible. So we set $\lambda_1$ to 50. Furthermore, we observe that our DSDM improves by 2.3% over the suboptimal M3D result despite $\lambda_1 = 0.01$, which further illustrates the importance of semantic diversity matching in dataset distillation.

**Effect of $\lambda_2$.** Similarly, we observe the effect of the weighting parameter $\lambda_2$ on the performance of the historical prototype. When $\lambda_2 = 0.2$, our DSDM performance reaches the optimal value. As $\lambda_2$ continues to increase, the performance decreases, which may be because the integration of the historical model is quite different from the current model, causing the current model to deviate from the class center of the original data.

## 5.6 Visualization

**Distribution Visualization.** To qualitatively illustrate the effectiveness of our DSDM in modeling the distribution of the original dataset, we use t-SNE to visualize the distilled data of the same category on CIFAR-10 for IPC=10/50 obtained by DM and DSDM, respectively. The results are depicted in Figure 7, where the "green dots" and the "red stars" stand for the real data and distilled data. It is obvious that the distilled data obtained by DSDM distribute more dispersedly than those of DM, especially for the distilled instances in the blue circles.

**Distilled Images.** To evaluate whether our method captures the semantic diversity of the original dataset, we visualize the distilled data of DM and our DSDM on CIFAR-10 in Figure 8. Each class contains 10 images. Our observations lead to two key conclusions: (1) The distilled data of our DSDM are more different compared to those of DM, proving that our covariance matrix alignment captures the semantic diversity of the original dataset. (2) The distilled data of our DSDM is more discriminative between each class, indicating

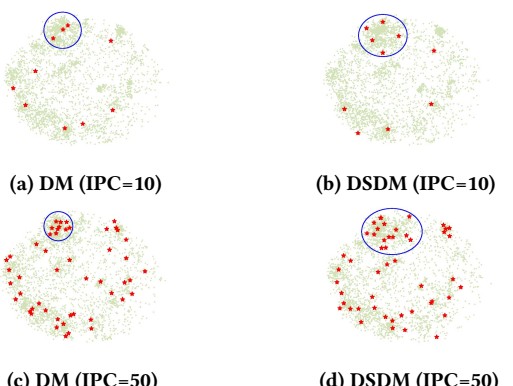

(a) DM (IPC=10)   (b) DSDM (IPC=10)

(c) DM (IPC=50)   (d) DSDM (IPC=50)

Figure 7: The data distributions of original images and distilled images through DM and DSDM for the same category on CIFAR-10, with IPC=10/50. "Green dots" denote the real data, while "red stars" represent the distilled data.

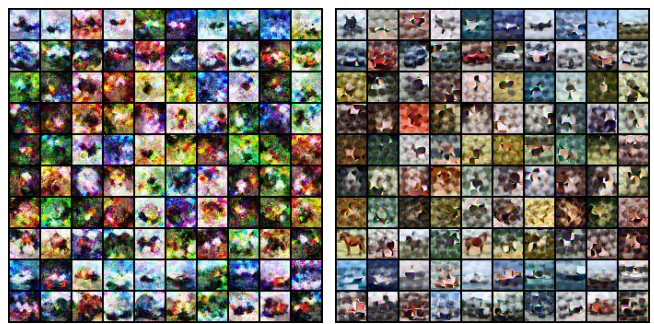

(a) The distilled images of DM.   (b) The distilled images of DSDM.

Figure 8: Visualizations of distilled images generated by DM and DSDM on CIFAR-10 with IPC=10.

that the class center of the pre-trained model effectively preserves the inherent features of the classes.

## 6 CONCLUSION

In this work, we propose a novel approach called Diversified Semantic Distribution Matching (DSDM) for Dataset Distillation, which explicitly aims to capture the semantic diversity features of the original dataset. DSDM consists of three carefully designed losses: prototype alignment, semantic diversity alignment, and historical prototype alignment. The prototype alignment and semantic diversity alignment modules are employed to capture the distribution of the original dataset. The historical prototype alignment enables DSDM to perform more stably during the training process. Experimental results across five image datasets and one speech dataset demonstrate that our DSDM outperforms existing techniques and generalizes well across different model architectures. In the future, we will explore how to better capture semantic diversity in more challenging scenarios of a low condensation ratio (e.g., IPC=1).

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
