# OpenReview forum: "Diversified Semantic Distribution Matching for Dataset Distillation"
_acmmm.org/ACMMM/2024/Conference — MM2024 Oral_

### Official Review · Reviewer_EiAw · 2024-05-09

**Rating:** 4
**Confidence:** 3

**Summary:**

This paper proposes a diversified semantic distribution matching method for dataset distillation. Specifically, the distribution matching is conducted on both prototype embeddings and covariance matrix reflecting the sample relationship. The proposed method achieves state-of-the-art performance on multiple dataset distillation benchmarks. The covariance matrix matching significantly improves the validation performance, supporting its effectiveness.

**Strengths:**

1. The idea of matching covariance matrix further enhances the potential of distribution matching for DD.
2. The writing is generally easy to follow.
3. The effectiveness of the proposed method is well supported by the experimental results.

**Limitations:**

1. In terms of diversity matching, a more intuitive idea would be matching the feature variance. Can authors explain why covariance is used here instead of variance? Are there theoretical or empirical analysis supporting the choice?
2. Authors claim that when only matching prototypes, "the algorithm may lazily minimize the alignment loss by distilling instances near the center of each class into the distilled data". However, in Fig. 7, the naive DM actually yields a distilled dataset that well covers the whole distribution instead of clustered at the class center.
3. The instability caused by different model architectures is not necessarily harmful. And aligning the historical prototypes doesn't bring obvious performance improvement in Tab. 4. The contribution of this part requires further evidence.
4. Minor:
- Line 364. Why "sub-network"?
- Line 368. "matries" -> "matrices"
- Reference not sufficient. More recent works should be investigated.

**Suitability:**

3

---

### Official Review · Reviewer_fWJR · 2024-05-24

**Rating:** 4
**Confidence:** 3

**Summary:**

In this paper, authors propose a Diversified Semantic Distribution Matching framework for efficient dataset condensation. Results on benchmarks prove the effectiveness or proposed method.

**Strengths:**

1.The written is clear and easy to follow.

2.The expansion to semi-supervised setting is reasonable and results seem promising.

3.high completion quality.

**Limitations:**

1. The method and model performance rely heavily on a large number of pretrained models, which significantly hinders the method's effectiveness. The feasibility of the current method is based on small-scale datasets like CIFAR-10/100. If the scale of data is increased to a very high level, such as millions of images, the cost of pre-training models becomes non-negligible. IDM also has this problem. Could the authors provide some theoretical or empirical insights and discussions on the relationship between the number of pre-trained models and the scale of the dataset? The results in Figure 5 are not solid.


2. Have the authors considered the scalability limit of the dataset distillation task? Assuming, it is unreasonable to compress an image dataset with 10,000 classes, millions of images, and a resolution of 3 \* 224 \* 224 into IPC=1, 10, 50. Then, if we suppose each class is to be compressed into 1000 synthetic samples, our optimization target becomes a huge 5D matrix of 10,000 \* 1000 \* 3 \* 224 \* 224, which will significantly increase resource consumption and computational costs, even if compression is done class by class. Could the authors further discuss the advantages and disadvantages between dataset distillation and dataset selection, and whether the designed method could be applied to dataset selection? Of course, this point does not seriously constitute a limitation of this article, but I would like to see the authors' discussion and thoughts on this issue.

**Suitability:**

2

---

### Official Review · Reviewer_ANHG · 2024-05-30

**Rating:** 4
**Confidence:** 3

**Summary:**

This paper upgrades the classic distribution matching method in Dataset Distillation by matching the prototype and covariance matrix. For better stability of training, the authors introduce the memory bank to save history synthetic data. The results on multiple datasets show that the new method outperforms several competitors. Ablation study proves that the upgraded distribution matching suppresses the classic one.

**Strengths:**

1. The paper is well-written and easy to follow.
2. The proposed method is well motivated. The memory bank is well-designed and can improve the training stability.
3. Experimental results are competitive to similar methods. Ablation study on the proposed modules has been provided and well justified. Hyper-parameters have been examed.

**Limitations:**

1. The paper lacks justification why no comparison to some popular methods, such as MTT.
2. The setting and details of the experiments on speech data are unclear.

**Suitability:**

2

---

### Official Review · Reviewer_fih9 · 2024-06-01

**Rating:** 4
**Confidence:** 3

**Summary:**

This paper proposes a novel dataset distillation method named Diversified Semantic Distribution Matching (DSDM). This method aims to enhance the semantic diversity and generalization capability of distilled datasets by aligning the semantic distributions of the original and distilled datasets. The experimental results across multiple image and speech datasets are great.

**Strengths:**

1. DSDM employs pre-trained models to extract semantic embeddings and models each class as a Gaussian distribution, which is novel in DM methods.
2. This matching strategy circumvents the complexity of bi-level optimization and ensures the distilled data maintains a high degree of semantic richness.
2. The experimental results across multiple image and speech datasets demonstrate superior generalization capabilities in various scenarios, highlighting its potential for widespread adoption in different domains​​.

**Limitations:**

1. The motivation and key component of this paper should be capturing the semantic features. However, the factor technology plays a greater role in improving performance. This undermines the paper's motivation and the method's validity.
2. I wonder how to set the value of the two hyperparameters 𝜆1 and 𝜆2.
3. The reliance on multiple pre-trained models to extract semantic embeddings adds to the complexity of the implementation.
4. This paper does not compare the time complexity with other DM methods. I would like to know how much additional time the DSDM method requires compared to DM, (e.g., calculating the covariance matrix).
5. If all the pre-trained models used are ConvNet-3? Have different architectures been tried to compose the model pool?

**Suitability:**

3

---

### Meta-Review · Area_Chair_8Ccw · 2024-06-24

**Recommendation:** Accept (Oral)
**Confidence:** 5

**Metareview:**

The paper proposes a novel dataset distillation method named Diversified Semantic Distribution Matching (DSDM). This method aims to enhance the semantic diversity and generalization capability of distilled datasets by aligning the semantic distributions of the original and distilled datasets. The experimental results across multiple image and speech datasets demonstrate the method's effectiveness, achieving state-of-the-art results.

Pros:
+ The paper is well-organized and easy to follow.
+ The use of covariance matrix matching to enhance the semantic diversity of distilled datasets is a novel and effective contribution.
+ The method achieves state-of-the-art results on multiple benchmarks, demonstrating its potential for widespread adoption in different domains.

Cons:
+ Some details have not been well solved according to the reviewers.

Given the 4 positive reviews and it may be the first dataset distillation work in ACM MM, I recommend accepting this paper for an Oral presentation.